## Protocol for a controlled human infection with genetically modified *Neisseria lactamica* expressing the meningococcal vaccine antigen NadA: a potent new technique for experimental medicine

Diane Gbesemete,[1,2] Jay Robert Laver,[3] Hans de Graaf,[1,2] Muktar Ibrahim,[3] Andrew Vaughan,[3] Saul Faust,[1] Andrew Gorringe,[4] Robert Charles Read[3,5]

For numbered affiliations see end of article.

**Correspondence to**
Dr Diane Gbesemete;
D.Gbesemete@soton.ac.uk

## ABSTRACT

**Introduction** *Neisseria lactamica* is a commensal organism found in the human nasopharynx and is closely related to the pathogen *N. meningitidis* (meningococcus). Carriage of *N. lactamica* is associated with reduced meningococcal carriage and disease. We summarise an ethically approved protocol for an experimental human challenge study using a genetically modified strain of *N. lactamica* that expresses the meningococcal antigen NadA. We aim to develop a model to study the role of specific bacterial antigens in nasopharyngeal carriage and immunity, to evaluate vaccines for their efficacy in preventing colonisation and to provide a proof of principle for the development of bacterial medicines.

**Methods and analysis** Healthy adult volunteers aged 18–45 years will receive an intranasal inoculation of either the NadA containing strain of *N. lactamica* or a genetically modified, but wild-type equivalent control strain. These challenge volunteers will be admitted for 4.5 days observation following inoculation and will then be discharged with strict infection control rules. Bedroom contacts of the challenge volunteers will also be enrolled as contact volunteers. Safety, colonisation, shedding, transmission and immunogenicity will be assessed over 90 days after which carriage will be terminated with antibiotic eradication therapy.

**Ethics and dissemination** This study has been approved by the Department for Environment, Food and Rural Affairs and South Central Oxford A Research Ethics Committee (reference: 18/SC/0133). Findings will be published in peer-reviewed open-access journals as soon as possible.

**Trial registration number** NCT03630250; Pre-results.

## INTRODUCTION

A controlled human infection experiment with a genetically modified (GM) *Neisseria lactamica* strain is currently underway. In the protocol presented here, organisms are inoculated into the nasopharynx of healthy volunteers to study the immune response to the modified organisms expressing the gene of interest. Volunteers, colonised with the strain harboured in the nasopharynx, will be allowed to leave the Clinical Research Facility (CRF) after a 4.5-day period of observation. This implies deliberate release of a genetically modified organism (GMO) so the protocol has been reviewed and approved by the UK Department for Environment, Food and Rural Affairs (DEFRA).[1]

*N. lactamica* and *N. meningitidis* are Gram-negative diplococci which both colonise the human nasopharynx. *N. lactamica* is non-pathogenic, non-encapsulated and lactose fermenting and is

a common commensal, particularly in young children.[2] [3] In contrast, *N. meningitidis* expresses polysaccharide capsule and although it usually colonises asymptomatically, it can in a minority of colonised individuals cause invasive disease.[4] [5] Due to recombination events, the organism exists in multiple clonal forms, with specific clonal complexes being characteristically associated with invasive disease.[6] Invasive meningococcal disease remains a significant global cause of morbidity and mortality with sporadic disease and small outbreaks throughout the world and significant epidemics occurring in the meningococcal belt of sub-Saharan Africa.[7]

### Strengths and limitations of this study

► This human challenge study using a genetically modified organism will provide insight into the role of a specific bacterial antigen in nasopharyngeal carriage and immunity, and provide a novel means to test the herd-immunity potential of vaccines.

► Safety is the first priority and has been considered at all points of the study design with extensive preclinical testing, a period of admission for close observation following inoculation and stringent infection control rules throughout the study.

► The use of environmental sampling and regular contact volunteer sampling will provide new information regarding the shedding and transmission of respiratory tract organisms.

► The planned inoculum dose is based on previous studies with wild-type *N. lactamica* and may not be the optimal dose to achieve colonisation with the genetically modified strains.

► The low number of participants may be insufficient to prove an effect of the expression of NadA on colonisation so further research may be required.

### Carriage of *N. lactamica* and *N. meningitidis*
Of note, *N. lactamica* appears to provide commensal-related protection against meningococcal disease. Age-specific rates of *N. meningitidis* carriage and disease are inversely proportional to carriage of *N. lactamica*.[8–10] The highest rate of natural carriage of *N. lactamica* occurs in infants. This then wanes in toddlers and older children and by adolescence carriage is approximately 1%.[2] [8] Carriage of *N. meningitidis* is low in infants, increasing gradually throughout childhood and peaking in adolescence with the highest rates of carriage seen in teenagers and university students.[11]

The mechanism of this epidemiological relationship is as yet undetermined. It is probably not due to cross-protective antibody production; the early years of life associated with high rates of *N. lactamica* carriage predate the development of natural bactericidal meningococcal antibodies.[4] Other postulated mechanisms include microbial competition, innate immune responses triggered by *N. lactamica* colonisation and cross-reactive non-humoral acquired immunity.[12] [13]

### Human challenge with *N. lactamica*
A controlled human infection model of *N. lactamica* colonisation has been used to investigate the mechanism of this natural effect. Previous studies have shown that human challenge with wild-type *N. lactamica* is safe and can induce long-standing colonisation. Over 350 healthy adult volunteers have been experimentally nasally inoculated with wild-type *N. lactamica* in previous studies. The colonisation fraction (the percentage of individuals who are colonised after challenge) was 35%–65%.[12] [13] Colonisation resulted in the development of humoral immunity to *N. lactamica* but no evidence of cross-reactive bactericidal antibodies to *N. meningitidis*. Some cross-reactive opsonophagocytic antibody production occurred but was rather weak.[13] In another large study, successful colonisation with *N. lactamica* was associated with the displacement of pre-existing meningococcal carriage, and inhibition of acquisition of *N. meningitidis*[12] supporting the role of *N. lactamica* carriage in protection from meningococcal carriage and therefore disease.

### Meningococcal vaccines
Glycoconjugate vaccines directed against capsular antigens for serogroups C, A, W-135 and Y have been in use globally for several years. These have had dramatic effects on disease incidence, which is probably mostly due to herd protection conferred by vaccine-induced modification of colonisation reducing interhost transmission.[14] [15] Recent vaccine developments include a new subcapsular vaccine, 4CMenB (Bexsero), which induces bactericidal antibodies against a range of strains, including serogroup B, and protects vaccinated infants against disease.[16] In view of the importance of carriage reduction for herd immunity, a large prospective randomised study was done to measure this, but the effect of Bexsero on carriage of *N. meningitidis* was found to be relatively modest and delayed until 3 months after vaccination,[17] with no evidence of an effect on carriage of the serogroup B organisms carried by the participants.

More rapidly effective and longer lasting vaccines are required, particularly to halt transmission during epidemics in the meningitis belt of sub-Saharan Africa. Successful future vaccines should maximise herd immunity by targeting carriage and transmission. The development of such vaccines requires a greater understanding of mucosal immune mechanisms and the specific antigens involved in colonisation.

### The meningococcal antigen NadA
In this human challenge study volunteers will receive intranasal inoculation with a GM strain of *N. lactamica* expressing the meningococcal antigen NadA. This antigen is being used because it is well defined, and one of the four strongly immunogenic components of the Bexsero vaccine. Bexsero has been demonstrated to be immunogenic in terms of generating serum bactericidal antibodies (SBA) against *N. meningitidis* strains that express NadA[18] and moderately effective in reducing acquisition of nasopharyngeal carriage of *N. meningitidis* over the course of 12 months after vaccination.[17] NadA expression by *N. lactamica* may induce systemic and mucosal immunity to NadA. When studied alongside a control strain, use of a GMO *N. lactamica* expressing NadA could permit advanced study of the mechanisms underlying mucosal immunity and carriage reduction. Furthermore, a GMO *N. lactamica* expressing NadA might exhibit enhanced protection against carriage of virulent *N. meningitidis*.

### Rationale for this study
The rationale for this study is to pilot the use of the transformed commensal *N. lactamica* as an experimental medicine tool to study immunity to meningococcal antigens in humans, and to investigate the potential utility of genetically transformed commensals as tools to investigate the efficacy of vaccines to prevent colonisation of organisms expressing specific antigens. Finally, expression of NadA might lead to increased efficiency of harmless

colonisation by *N. lactamica* and prompt the development of this GMO as a bacterial medicine.

## METHODS AND ANALYSIS
### Study overview
This is a prospective controlled human challenge study in which challenge volunteers will be inoculated intranasally with *N. lactamica* (recipient strain Y92-1009) genetically modified to express NadA (the intervention strain) or a control GM strain. An inoculum dose of $10^5$ colony-forming units will be used for both strains. Following inoculation, challenge volunteers will be admitted to Southampton National Institute for Health Research (NIHR) CRF for 4.5 days. A further group of volunteers, who are close contacts of the participants, will be enrolled to detect transmission of the inoculated strains. Safety parameters, colonisation, shedding, transmission and immunogenicity will be assessed during the admission period and over a follow-up period of approximately 3 months. Colonisation will be terminated with antibiotic eradication therapy on day 90, for all challenge and contact volunteers. The planned study period is from May 2018 to May 2020.

### Study objectives
The objectives of this study are to establish the safety and NadA-specific immunogenicity of nasal inoculation with the intervention strain of GM *N. lactamica* and to assess subsequent shedding and transmission. A further objective is to assess the efficacy of ciprofloxacin eradication therapy. These objectives and the study endpoints are summarised in table 1.

### GM *N. lactamica*
#### The intervention strain
The intervention strain (*N. lactamica* strain Y92-1009) has been transformed by the integration of the *N. meningitidis* gene *nadA* (NEIS1969), leading to expression of NadA. The NadA protein is a member of the type V autotransporter family of outer membrane proteins, and in *N. meningitidis* is associated with an increased level of adhesion to and invasion of human epithelial cell lines. The inserted gene is derived from *N. meningitidis* strain MC58, which contains *nadA* allele 1. The presence of the *nadA* gene in the genome is associated with hypervirulent lineages of *N. meningitidis*, but NadA surface expression has not been shown to be causal for increased virulence. Detailed molecular microbiological information can be found within the published DEFRA approval notice.[1]

#### The control strain
The control strain has been genetically modified in exactly the same way as the intervention strain, except that it does not contain the coding sequence for the *nadA* gene. In terms of gene content and behaviour in the laboratory, this strain is extremely similar to wild type. Using this strain as a control inoculum is superior to using the wild-type strain as the changes made to the genetic architecture and gene regulation are identical to the intervention strain apart from the insertion of *nadA*.

| Table 1 | Objectives and endpoints | |
|---|---|---|
| | **Objectives** | **Endpoints** |
| Coprimary objectives | To establish the safety of nasal inoculation of healthy volunteers with a genetically modified strain of *Neisseria lactamica* expressing NadA | Occurrence of unsolicited adverse events within the study period |
| | | Occurrence of serious adverse events within the study period |
| | To assess the NadA-specific immunity in healthy volunteers following nasal inoculation with *N. lactamica* expressing NadA | Rise in serological specific IgG titre (anti-NadA) comparing day 0 vs days 14–90 and comparing volunteers colonised by one of the two GMOs |
| | | Rise in mucosal specific antibody titre comparing day −5 vs days 3–90 and comparing volunteers colonised with the two GMOs |
| | | Change in nasal cytokine profile comparing day 0 vs days 3–90 and comparing volunteers colonised with the two GMOs |
| Secondary objectives | To assess the shedding of genetically modified *N. lactamica* following nasal inoculation | Culture of GM *N. lactamica* from environmental samples—comparing intervention and control groups |
| | To assess the transmission of genetically modified *N. lactamica* to bedroom contacts of inoculated volunteers | Culture of GM *N. lactamica* from throat swabs taken from contact volunteers from day 4 until day 90—comparing intervention and control groups |
| | To assess the efficacy of a single dose of ciprofloxacin in eradicating carriage of genetically modified *N. lactamica* | Culture of GM *N. lactamica* from throat swabs taken at the eradication visit in comparison to posteradication visit in challenge and contact volunteers |

GM, genetically modified; GMO, genetically modified organism.

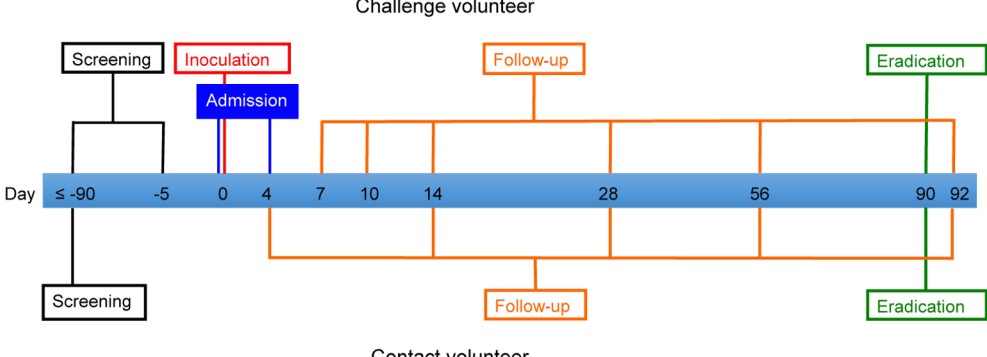

**Figure 1** Study timeline.

## Preclinical safety data

Both strains have been demonstrated to remain acutely susceptible to killing by normal human serum and retain sensitivity to the antibiotics used clinically to treat meningococcal disease (rifampicin, ciprofloxacin and ceftriaxone). Preclinical testing[1] has shown that the NadA autotransporter is functionally expressed in the intervention strain, the NadA protein is strongly immunogenic in the context of expression in *N. lactamica* and that the expression of NadA does not significantly increase the pathogenicity of the commensal in a murine model of infection. Neither strain has an increased propensity to become transformed by exogenous sources of DNA, which might otherwise allow it to acquire virulence factors such as an extracellular capsule, as compared with the wild-type strain.

## Quality assessment and control

Preparation, storage and monitoring of the challenge strains will be carried out to Good Manufacturing Practice (GMP)-like standards at the University of Southampton. The dose and purity of the inoculum will be determined after inoculation for quality assessment.

## The inoculum dose

Based on the previous *N. lactamica* human challenge studies it is estimated that 50% of volunteers will be colonised 1–2 weeks after inoculation at this dose.[13] Fifty per cent has been chosen as an acceptable colonisation rate because it is below a 'saturating' dose and therefore avoids the difficulties of interpretation of a challenge dose that is much higher than physiologically appropriate.

## Study volunteers
### Challenge volunteers

Healthy volunteers aged 18–45 years will be recruited and challenged until 11 volunteers in each group are colonised with GM *N. lactamica* at day 14 or up to a maximum of 22 inoculated volunteers in each group.

### Contact volunteers

Contact volunteers are bedroom contacts of challenge volunteers, defined as individuals who share a bedroom on at least one occasion during the study period. A maximum of one contact volunteer may be recruited

per challenge volunteer and contact volunteers must give informed consent prior to inoculation of the corresponding challenge volunteer. Bedroom contacts who are under 18 or who are immunocompromised will be excluded from participation, as will their corresponding challenge volunteer.

## Eligibility criteria

We will not recruit from vulnerable groups such as those with impaired capacity. Those with close contact with potentially vulnerable people such as small children and immunocompromised individuals will be excluded. Specific inclusion and exclusion criteria can be found in online supplementary table 1.

## Infection control agreement

Both challenge and contact volunteers must provide written infection control agreement prior to enrolment, which will include agreement to have no other bedroom contacts during the study period. Details of the infection control requirements can be found in the online supplementary table 2.

## Study setting

The challenge procedure, admission and follow-up visits will take place in the NIHR CRF at University Hospital Southampton NHS Foundation Trust.

## Recruitment

Participants will be recruited via a variety of media including ethically approved adverts displayed within the hospital, on Southampton NIHR CRF websites, social media and circulated literature, the Southampton CRF database of healthy volunteers, presentations and press releases. Individuals who express an interest will be sent a volunteer information sheet. Volunteers will be offered reimbursement for their time, travel and inconvenience.

## Study timeline

Challenge and contact volunteers will be enrolled from the date of screening, up to 90 days prior to the challenge procedure, until day 92 after challenge. The duration of volunteer participation will therefore be up to approximately 6 months. An overview of the study timeline is shown in figure 1. Details of study procedures are shown

in online supplementary table 3 (challenge volunteers) and online supplementary table 4 (contact volunteers).

## Screening

Potential challenge and contact volunteers will be invited to separate screening visits up to 90 days prior to challenge. At these screening visits they will be fully informed of all aspects of their involvement in the study, be given an opportunity to ask questions, to give informed consent and to undergo a medical screening to determine eligibility. Challenge volunteers will be asked to complete a preconsent questionnaire to ensure their understanding of the study and their medical history will be confirmed with their general practitioners. The infection control guidelines (see online supplementary table 2) will be explained to all volunteers and they will be asked to complete a questionnaire to confirm their understanding of these guidelines, and to sign an agreement to follow these guidelines throughout the study period. Challenge volunteers will attend a prechallenge visit the week prior to their challenge to ensure that they remain eligible.

## First volunteers

For each GM strain the first volunteers will be challenged individually and then in pairs with a safety review after volunteers 1, 3 and 5. Further volunteers will be challenged in groups of a maximum of 5.

## Challenge

Challenge volunteers will be admitted to a designated area of the NIHR CRF on the morning of their challenge procedure. Ongoing informed consent and eligibility will be confirmed and clinical samples will be taken for baseline immunology.

The inoculum will be prepared from frozen stocks and will be administered by a study doctor following study-specific standard operating procedures. The challenge will take place in an environmental chamber within the CRF. The challenge volunteer will be positioned supine with neck extended and breathing normally through their mouth. 0.5 mL of inoculum will be administered slowly from a pipette into each nostril. The residual inoculum will be analysed to confirm the administered dose and purity. Public Health Southampton will be informed of all participants who have been challenged with the GMOs.

## Admission

During admission, challenge volunteers will have access to an individual bedroom, shared bathroom facilities and a shared recreational area. Clinical observations and any symptoms will be recorded approximately every 4 hours and a study doctor will review volunteers twice a day. Clinical and environmental samples will be taken as detailed in table 2 to assess safety, colonisation, immunogenicity and shedding.

Prior to discharge of the challenge volunteer, the contact volunteer will attend to confirm ongoing consent and eligibility and the infection control procedures will be reiterated to both challenge and contact volunteers.

| Table 2 | Study procedures during admission | | | | |
|---|---|---|---|---|---|
| | Day 0 | Day 1 | Day 2 | Day 3 | Day 4 |
| Vital signs | Preinoculation then four hourly | Four hourly | Four hourly | Four hourly | Four hourly |
| Review of adverse events | Four hourly | Four hourly | Four hourly | Four hourly | Four hourly |
| Medical review | ×2 | ×2 | ×2 | ×2 | ×2 |
| Pregnancy test (females only) | + | | | | |
| Review eligibility | + | | | | |
| Inoculation | + | | | | |
| Throat swab (culture) | + | | + | + | + |
| Throat swab (microbiome) | + | | | + | |
| Nasal wash | | | | + | |
| Nasosorption test | + | | | + | |
| Saliva sample | + | | | + | |
| Environmental samples | | + | + | + | + |
| Safety bloods (mL) | 8 | | | 8 | |
| Immunological blood tests (mL) | 70 | | | | |

## Follow-up

Following challenge volunteer discharge, volunteers will be monitored for adverse events, colonisation, shedding, transmission and immunogenicity as detailed in the online supplementary table 3 (challenge volunteers) and online supplementary table 4 (contact volunteers).

### Adverse events

Adverse events will be monitored at each follow-up visit. In addition to this volunteers will be encouraged to contact the study team at any point during the study in the event any symptoms develop.

### Colonisation

Colonisation will be assessed by culture of throat swabs and nasal washes. Colonisation density will be estimated by qPCR and comparison will be made between the intervention and control groups.

### Shedding

Shedding of GM *N. lactamica* from inoculated challenge volunteers will be assessed by microbiological analysis of environmental samples. Comparison of shedding will be made between the intervention and control challenge volunteers. Environmental sampling will include culture and PCR of face mask samples and air samples taken within an environmental chamber during aerosol-producing activities.

A challenge volunteer in the intervention group will be considered to have increased shedding at a particular time point if they have a 10-fold increase in shedding in comparison to the average shedding seen at the same time point in colonised control group volunteers to date. This is a nominal figure agreed with the statutory authority (UK DEFRA) because of the unpredictable scale and frequency of this event which will not permit a prospective, statistically based assessment of potentially hazardous release to the environment. If increased shedding is seen at any point from the day 14 visit then the volunteer will be asked to attend as soon as possible for an additional shedding check visit. If increased shedding is seen at two consecutive visits this will be considered enhanced shedding.

### Transmission

Transmission will be assessed by culture and PCR of throat swabs from contact volunteers. Comparison will be made between the intervention and control groups.

### Immunogenicity

Mucosal and systemic immunogenicity will be investigated. Saliva and nasal secretions will be collected for assessment of mucosal immunogenicity and blood samples for systemic humoral and cellular responses.

## Eradication

Antibiotic eradication therapy will be given to all challenge and contact volunteers with a throat swab to confirm successful eradication after a maximum of 48 hours. Standard eradication will be given to all volunteers at day 90 (regardless of colonisation status) with a confirmatory throat swab on day 92. Eradication therapy may be given at an earlier time point under specific circumstances.

Triggered eradication may be given to volunteers at any time point due to:
► Safety concerns in the challenge volunteer or corresponding contact volunteer, at the discretion of the study team.
► Enhanced shedding from the challenge volunteer.
► Study withdrawal for any other reason.

If eradication is triggered for a challenge or contact volunteer then their corresponding challenge or contact volunteer (if applicable) will receive eradication therapy on the same day and both volunteers will be withdrawn from the study.

In addition to this, contact volunteers found to be colonised with GM *N. lactamica* at any point may receive early eradication therapy, as ongoing colonisation of contact volunteers is not required to fulfil the study objectives. In this case, the corresponding challenge volunteer will not receive eradication therapy and both will continue in the study as planned.

A single dose of 500 mg ciprofloxacin will be taken under supervision of the study team. All female volunteers will have a pregnancy test prior to eradication. In the event of a positive pregnancy test, alternative eradication therapy will be used—rifampicin 600 mg twice daily for 48 hours.

Both rifampicin and ciprofloxacin, as oral antibiotics, have been shown to be effective in eradicating carriage of *N. meningitidis*,[19] and are regularly used as postexposure prophylaxis.[20] Both GM strains are also sensitive to these antibiotics.

### Study holding rules

An independent external safety committee will review the safety aspects of the study on a regular basis and in the event of any significant safety concerns. Colonisation, shedding, transmission and clinical parameters will be closely monitored throughout the study. In the event of a study holding criterion being met the study will be paused for a safety review. No further volunteers will be challenged until the data have been reviewed by the external safety committee and study continuation approved.

### Enhanced colonisation

The expression of NadA by the intervention strain of GM *N. lactamica* is expected to be associated with either an increase or a decrease in colonisation frequency or density compared with wild type. Colonisation rate and density estimation will be monitored but an increase in colonisation alone will not trigger a study pause unless associated with sustained enhanced shedding, transmission or safety concerns.

### Enhanced shedding

Enhanced shedding triggering early eradication in three or more of the first five volunteers to receive the intervention strain or in >50% of ongoing challenge volunteers in the intervention group will trigger a study pause.

### Enhanced transmission

Transmission of either strain of GM *N. lactamica* to three of the first five or >50% of ongoing contact volunteers will trigger a study pause.

### GM N. lactamica disease

If antibiotic treatment (intravenous ceftriaxone or intravenous chloramphenicol) is given to any volunteer due to possible GM *N. lactamica* disease then a study pause will be triggered.

## Sample size

We are aiming to achieve colonisation in 10 challenge volunteers for each of the GM strains. This is based on a previous experimental *N. lactamica* challenge study, which showed a significant rise in serological antibody titre against *N. lactamica* over 2 weeks.[13] This gave SDs on a log-10 scale of 0.11 for IgA saliva and 0.26 for serum total IgG. For this study, using the SD of 0.26 we will be able to confirm a fourfold rise of anti-NadA with 10 carriers of *N. lactamica* expressing NadA with 90% power using analysis of variance.

Allowing for a dropout rate of approximately 10%, we will therefore recruit challenge volunteers until we have 11 individuals colonised for each group up to a maximum of 22 volunteers for each group. Estimating a colonisation fraction of 50%, approximately 44 individuals will be enrolled as challenge volunteers. A maximum of one contact volunteer will be enrolled per challenge volunteer.

## Patient and public involvement

A patient and public involvement (PPI) group was consulted during the early stages of study design to discuss the implications of human challenge with a GMO. An important suggestion arising from this consultation was to seek information about the potential for spread of infection which we have discussed further with Public Health England experts and DEFRA. As a result of these discussions, our protocol includes close monitoring of environmental shedding and transmission to sleeping partners with specific action points in the event that there is evidence of enhanced shedding into the environment. Suggestions from the PPI consultation were also used in the design of the volunteer information sheet.

In addition, formal and informal feedback from volunteers involved in other human challenge trials in the NIHR CRF Southampton has been used to refine the design of this study and preparation of the admission area.

Participants in this study will be provided with a lay summary of the results once available.

## ETHICS AND DISSEMINATION

As this study involves the deliberate release of GM bacteria into the community it has been considered and approved by the responsible government ministry—the DEFRA.[1]

Results will be published in peer-reviewed journals once available.

## DISCUSSION

### Human challenge with a GMO: safety considerations

This study will result in the deliberate release of two GMOs. One previous study has been published in which volunteers were deliberately inoculated with a GMO that has therefore potentially been released into the general population. In that study, carried out in Sweden, a GM attenuated *Bordetella pertussis* strain was constructed as a vaccine candidate. This was administered nasally, in order to mimic natural infection without inducing disease and volunteers were subsequently followed up as outpatients.[21]

In the UK the deliberate release of a GMO requires DEFRA approval. This protocol has therefore been reviewed by DEFRA who have considered the potential for colonisation of other members of the general population, and have given approval of the study.

During the design of this study, our priority has been to ensure the safety of the volunteers to limit the potential for transmission to close contacts of the volunteers, study team members and the wider population. A number of safety considerations have been incorporated into the protocol and an independent external safety committee will review the safety aspects of the study on a regular basis.

### Safety of GM N. lactamica

*N. lactamica* is a non-virulent commensal organism and there have been no safety concerns in previous challenge studies with the wild-type organism. There is no evidence to suggest that the GM strains will be more likely than wild type to cause invasive disease, as the organisms are non-capsulate and highly susceptible to killing by human serum. Preclinical work has indicated that the GMOs are stable, do not undergo recombination events at higher frequency than wild type and are non-virulent when inoculated into mice. We therefore consider that the likelihood of the GMO causing any disease is extremely low.

### Safety of challenge and contact volunteers

For each strain, the first five challenges will be staggered with a safety review between challenges. All challenged volunteers will be admitted to Southampton NIHR CRF for close observation for 4.5 days following challenge. The period of risk of development of invasive meningococcal disease is the first 48 hours following acquisition, so in the unlikely event of any volunteer developing symptoms it would be expected to occur within this period of admission. The NIHR CRF is funded and staffed to allow the delivery of higher risk experimental studies and is located within an NHS hospital so study nurses will be

immediately available, study doctors will be contactable and able to attend and full NHS clinical services will be present within the same building if required. Following discharge all volunteers will be monitored regularly for adverse events and will be given a 24-hour phone number to contact the study team.

### Minimising onward transmission

Transmission occurs through close contact and previous studies looking at the transmission of *N. lactamica* and *N. meningitidis* suggest that household members, and in particular bedroom sharers of colonised individuals, are those at highest risk of acquisition of carriage.[22–24] Bedroom sharers of challenge volunteers are therefore the most relevant community members to screen for transmission and so will give informed consent and will be enrolled as contact volunteers for this purpose. Potential challenge or contact volunteers with household members or other close contacts who may be at increased risk of acquisition of carriage or of *N. lactamica* disease will be excluded from the study.

Other infection control measures include the use of Personal protective equipment (PPE), strict infection control guidelines and close monitoring of shedding and transmission. These measures have been designed to limit the potential onward transmission of the inoculated bacteria to study team members, vulnerable individuals and to the general population. In addition all volunteers will receive eradication therapy prior to study completion, regardless of their colonisation status.

### The benefit of a human challenge model

A greater understanding of the mucosal immune mechanisms of protection from colonisation is essential for the development and evaluation of new vaccines, specifically ones targeting colonisation and transmission. The most direct and effective way to achieve this is experimental controlled human infection. This model can be used to investigate in detail components of mucosal and systemic immunity activated in real time following infection with a defined antigen. Also, this model could be used to investigate vaccine efficacy. For example, healthy volunteers who have received a study vaccine could then be challenged with a defined organism expressing constituent antigens. Monitoring carriage of the challenge bacterium over time would then provide information of the efficacy of the vaccine in the prevention of colonisation. Experimental human challenge with pathogens of interest such as *N. meningitidis* would be potentially hazardous and therefore raise significant ethical and logistical issues. The use of a harmless commensal organism that has been transformed to express specific antigens could be a safe and effective alternative.

*N. lactamica* is an appropriate organism to be transformed for this purpose. It is a well-studied and characterised commensal organism, which is known to exclusively colonise the human nasopharynx. It is genetically very similar to *N. meningitidis*, sharing approximately 67% of the genes believed to be associated with meningococcal virulence.[25] Despite this, *N. lactamica* is known to be non-virulent and has been used safely in previous human challenge studies.

*N. lactamica* is the only member of the genus *Neisseria* which is able to ferment lactose due to the activity of β-D-galactosidase coded for by the gene *lacZ*. This causes colonies to grow blue on the chromogenic substrate 5-bromo-4-chloro-3-indolyl β-D-galactopyranoside (X-gal). This characteristic has been used in our study; both of our GM strains have been derived from a *lacZ* deficient strain of *N. lactamica* Y92-1009 ($\Delta lacZ$), which grows as white colonies on X-gal-containing medium. During the transformation process *lacZ* has been reintegrated as a marker of successful transformation, thus allowing screening for successful transformants on the basis of blue/white colony formation on X-gal-containing medium. This has been done to completely avoid the use of genes coding for resistance to antibiotics and to eliminate the risk of our challenge experiment disseminating antimicrobial resistance genes into the nasopharyngeal microbiome.

The meningococcal antigen NadA has been chosen as the specific antigen for this study. NadA is a component of the Bexsero vaccine and is known to be potently immunogenic so successful colonisation is likely to induce the production of specific anti-NadA antibodies. Indeed, in a murine nasal challenge model, where GM *Streptococcus gordonii* expressing meningococcal NadA was used to inoculate mice, colonised subjects produced systemic anti-NadA bactericidal antibodies and localised anti-NadA IgA.[26] The *nadA* gene is associated with hypervirulent strains of *N. meningitidis* and was present in 50% of strains isolated from cases of meningococcal disease.[27] NadA has a role in increased adhesion and invasion into human epithelial cells[28] so NadA expression may therefore increase the ability of *N. lactamica* to colonise the nasopharynx. However, *nadA* is absent from some virulent strains and the majority of non-virulent strains of *N. meningitidis*, which may limit the potential for cross-reactive immunity.[27 29] In addition, as NadA is so potently immunogenic, expression may in fact reduce the duration of colonisation due to enhanced clearance.

Once this human challenge model has been shown to be safe and effective it could potentially be used to study other meningococcal antigens, or indeed antigens from other respiratory mucosal pathogens.

### The potential for use as a bacterial medicine

Carriage of wild-type *N. lactamica* appears to be protective against meningococcal disease, at least partly due to physical competition. The modification of *N. lactamica* to express an adhesion such as NadA could plausibly improve the colonisation fraction or colonisation duration.

Colonisation with *N. lactamica* has been shown to result in some cross-reactive acquired immunity to *N. meningitidis*, but this is insufficient to be fully protective.[13] Genetic modification of *N. lactamica* to express a meningococcal antigen known to be potently immunogenic may lead to the production of anti-meningococcal SBAs.

If successful, these improvements in the protective effect of induced colonisation with *N. lactamica* may lead to its potential use as a bacterial medicine.

## CONCLUSION

The successful and safe colonisation of healthy volunteers with GM strains of *N. lactamica* will pave the way for further challenge studies involving transformants which express other meningococcal antigens, and potentially antigens expressed by other pathogens. These challenge models will lead to a greater understanding of mucosal immune responses to colonisation and infection, provide a platform for the development and assessment of improved vaccines and may lead to the development of novel bacterial medicines.

**Author affiliations**
[1]NIHR Clinical Research Facility, University Hospital Southampton NHS Foundation Trust, Southampton, UK
[2]Faculty of Medicine, University of Southampton, Southampton, UK
[3]Clinical and Experimental Sciences, University of Southampton, Southampton, UK
[4]Research, Public Health England Porton, Salisbury, Wiltshire, UK
[5]NIHR Southampton Biomedical Research Centre, Southampton, UK

**Acknowledgements** We acknowledge the input of a Public and Patient Involvement group in the early design stage of this study.

**Contributors** The study was designed by DG, JRL and RCR with contributions from HdG, MI, AV, SF and AG. The first drafts of the manuscript were prepared by DG, JRL and RCR. HdG, MI, AV, SF and AG contributed to editing and approval of the final version of the manuscript.

**Funding** This work will be supported by the Medical Research Council (Grant MR/N026993/1) 'Pathfinder: Experimental Human Challenge with Genetically Modified Commensals to Investigate Respiratory Tract Mucosal Immunity and Colonisation' and the MRC Confidence in Concept Award, with additional funding from Experimental Medicine by the National Institute for Health Research through support from the Southampton NIHR CRF and the Biomedical Research Centre. The development of the technology underpinning the genetic modification was funded by the Medical Research Council (a genetically modified nasopharyngeal commensal as a platform for bacterial therapy, MR/N013204/1).

**Competing interests** JRL and RCR declare a potential conflict of interest: The patent WO2017103593-A1 'New modified *Neisseria lactamica* transformed with recombinant DNA encoding heterologous protein, used for e.g. prophylactic treatment of pathogenic infection, preferably meningococcal infection', is assigned to the University of Southampton, with JRL and RCR as inventors.

**Patient consent for publication** Not required.

**Ethics approval** It has also been reviewed and approved by South Central Oxford A Research Ethics Committee (SC/18/0113) and by the UK Health Research Authority (IRAS ID 235090).

**Provenance and peer review** Not commissioned; externally peer reviewed.

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
