## [Reviewer comments · BMJ Open]

ARTICLE DETAILS

TITLE (PROVISIONAL)	Protocol for a controlled human infection with genetically modified Neisseria lactamica expressing the meningococcal vaccine antigen NadA: A potent new technique for experimental medicine
AUTHORS	Gbesemete, Diane; Laver, Jay; de Graaf, Hans; Ibrahim, Muktar; Vaughan, Andrew; Faust, Saul; Gorringer, Andrew; Read, Robert

VERSION 1 - REVIEW

REVIEWER	Caroline Vipond National Institute of Biological Standards and Control
REVIEW RETURNED	23-Nov-2018

GENERAL COMMENTS	Summary The trial is to understand the safety of genetically modified Neisseria N. lactamica, which express the meningococcal antigen NadA given intranasally in humans. Safety will be assessed by close monitoring of the volunteers and their 'bedroom' contacts, looking for any unsolicited or serious adverse events. In addition, shedding of the organism from volunteers and their contacts will be measured and if found to be high antibiotics will be given to interrupt colonisation. Serological analysis to determine a rise in IgG against NadA will be measured, alongside a rise in mucosal antibody titre and changes in the nasal cytokine profile. The trial has been well-planned out and all ethical issues addressed. The group have run similar trials and I am confident they will be able to run this one safely and effectively. Comments: In a previous trial the group found that an IgG immune response was not raised against N. lactamica proteins when the organism was commensal in the nasopharynx of volunteers. The mechanism of protection offered by the N. lactamica against invasive meningococcal disease, which has been documented over many years, therefore remains undefined, but in the 2015 study was attributed to some other effector which resulted in the reduction of carriage of Neisseria meningitidis. The interruption of carriage of the Neisseria meningitidis has been shown to be a major factor in the success of glycoconjugate vaccines to prevent meningococcal disease caused by group C meningococci. In this trial the investigators have manipulated the commensal N. lactamica organism to express NadA, a vaccine antigen in Bexsero believed to play a role in adhesion. Where I think the scientific proposal gets a little confusing is what role they expect NadA to play. On one had the serological response to NadA, as
---

	measured by IgG is one of the primary objectives of the study, suggesting it has been added with a view to generating a response by the adaptive immune system, the protective route of other meningococcal vaccines. Whereas in line 46 the investigators say: 'expression of NadA might lead to increased efficiency of harmless colonisation by N. N. lactamica and prompt the development of this GMO as a bacterial medicine.' The latter would perhaps be a stronger argument for NadA inclusion as to date there is no evidence that a nasal dose of N. lactamica will give rise to an IgG response. The protocol also states that it might not be sufficiently powered to demonstrate the a response to NadA, however, if there is no evidence of any IgG against NadA in 10 donors following carriage of N. lactamica, I would suggest this would indeed add weight to the findings that N. lactamica does not reduce disease through the IgG and subsequent complement killing and that the investigators should focus on alternative mechanisms which appears to be important. I would therefore have had the mucosal antibody titres and nasal cytokine profiles as primary objectives as to date evidence suggests they are more important than the adaptive immune response.
--	--

REVIEWER	Jeremy Brown University College London
REVIEW RETURNED	30-Nov-2018

GENERAL COMMENTS	A well written and comprehensive description of a experimental infection in humans trial of a genetically modified N lactamica. My only suggestion is that in the description of the mutant they state what N. lactamica strain background has been used, as well as (if known) the site of insertion of the meningococcal gene
---

REVIEWER	Abad, Raquel ISCI, Spain
REVIEW RETURNED	07-Jan-2019

GENERAL COMMENTS	The authors present a study protocol for a planned research. The study has been approved by the DEFRA and detailed consent documents are available from https://www.gov.uk/government/publications/genetically-modified-organisms-university-of-southampton-17r5001#history (the link provided by the authors in reference 1 is wrong). The study design and methodology presented are appropriated, although several points would need to be clarified:  • According to the study protocols submission guidelines, the dates of the study should be included in the manuscript. • Due to the nadA gene variability, information about donor strain regarding to the nadA gene allele should be include. • N. lactamica or N. meningitidis detected on challenge volunteers throat swab or nasal wash taken at screening or at the pre-challenge visit is considered an exclusion criterion in the manuscript while in the study documents approved by DEFRA the exclusion criterion considered is pre-existing carriage of Neisseria spp (assessed 7 days prior to inoculation).
---

	• Why exclusion criteria 2, 5, 6 and 7 (Supplementary table 1) are only for challenge volunteers? The presence of these criteria in contact volunteers could affect their capability to acquisition of carriage and therefore transmission results obtained from the study. There are several references in the text not included in the "References" section (Harrison 2009, Evans 2011 . . .).
--	---

VERSION 1 – AUTHOR RESPONSE

Reviewer(s) Reports:

Reviewer: 1

Reviewer Name: Caroline Vipond

Institution and Country: National Institute of Biological Standards and Control

Please state any competing interests or state 'None declared': none declared

Please leave your comments for the authors below

See attached

Summary

The trial is to understand the safety of genetically modified *Neisseria N. lactamica*, which express the meningococcal antigen NadA given intranasally in humans. Safety will be assessed by close monitoring of the volunteers and their 'bedroom' contacts, looking for any unsolicited or serious adverse events. In addition, shedding of the organism from volunteers and their contacts will be measured and if found to be high antibiotics will be given to interrupt colonisation.

Serological analysis to determine a rise in IgG against NadA will be measured, alongside a rise in mucosal antibody titre and changes in the nasal cytokine profile.

The trial has been well-planned out and all ethical issues addressed. The group have run similar trials and I am confident they will be able to run this one safely and effectively.

Comments:

In a previous trial the group found that an IgG immune response was not raised against *N. lactamica* proteins when the organism was commensal in the nasopharynx of volunteers. The mechanism of protection offered by the *N. lactamica* against invasive meningococcal disease, which has been documented over many years, therefore remains undefined, but in the 2015 study was attributed to some other effector which resulted in the reduction of carriage of *Neisseria meningitidis*. The interruption of carriage of the *Neisseria meningitidis* has been shown to be a major factor in the success of glycoconjugate vaccines to prevent meningococcal disease caused by group C meningococci. In this trial the investigators have manipulated the commensal *N. lactamica* organism to express NadA, a vaccine antigen in Bexsero believed to play a role in adhesion. Where I think the scientific proposal gets a little confusing is what role they expect NadA to play. On one had the serological response to NadA, as measured by IgG is one of the primary objectives of the study, suggesting it has been added with a view to generating a response by the adaptive immune system, the protective route of other meningococcal vaccines. Whereas in line 46 the investigators say: 'expression of NadA might lead to increased efficiency of harmless colonisation by *N. N. lactamica*

and prompt the development of this GMO as a bacterial medicine.’ The latter would perhaps be a stronger argument for NadA inclusion as to date there is no evidence that a nasal dose of N. lactamica will give rise to an IgG response.

The protocol also states that it might not be sufficiently powered to demonstrate the a response to NadA, however, if there is no evidence of any IgG against NadA in 10 donors following carriage of N. lactamica, I would suggest this would indeed add weight to the findings that N. lactamica does not reduce disease through the IgG and subsequent complement killing and that the investigators should focus on alternative mechanisms which appears to be important. I would therefore have had the mucosal antibody titres and nasal cytokine profiles as primary objectives as to date evidence suggests they are more important than the adaptive immune response.

We thank Dr Vipond for these comments. We agree with her, but the serological outputs have been included as the primary outcome measure because they are the responses that we proposed to the funder (Medical Research Council). It may well be the case that (a) the transformed strain colonises but is not immunogenic (which will be interesting), (b) that the transformed strain is a much more efficient colonizer or (c) that the transformed strain is quickly eliminated by the host because of a powerful immune response. We will not know which of these is the case until we complete the study. Regarding mucosal responses, we are collecting appropriate samples to do this, and they are secondary outcome measures as stated.

Reviewer: 2

Reviewer Name: Jeremy Brown

Institution and Country: University College London

Please state any competing interests or state ‘None declared’: none declared

Please leave your comments for the authors below

A well written and comprehensive description of a experimental infection in humans trial of a genetically modified N lactamica. My only suggestion is that in the description of the mutant they state what N. lactamica strain background has been used, as well as (if known) the site of insertion of the meningococcal gene. – We thank Professor Brown. The background strain is Y92-1009, and we have stated that the detailed molecular microbiology can be found in the cited published DEFRA approval document.

Reviewer: 3

Reviewer Name: Raquel Abad

Institution and Country: ISCIII, Spain

Please state any competing interests or state ‘None declared’: None declared

Please leave your comments for the authors below

The authors present a study protocol for a planned research. The study has been approved by the DEFRA and detailed consent documents are available from <https://emea01.safelinks.protection.outlook.com/?url=https%3A%2F%2Fwww.gov.uk%2Fgovernment%2Fpublications%2Fgenetically-modified-organisms-university-of-southampton->

17r5001%23history&data=01%7C01%7CR.C.Read%40soton.ac.uk%7Cf12ab65f30bb453a00cd08d6797024d8%7C4a5378f929f44d3e8e89669d03ada9d8%7C1&sdata=aTYtGmL1Ca3LxYd0zsG%2BokaZG41yNEVO9pFAMrFk3s%3D&reserved=0 (the link provided by the authors in reference 1 is wrong). Thanks for this - we have corrected it.

The study design and methodology presented are appropriated, although several points would need to be clarified:

- According to the study protocols submission guidelines, the dates of the study should be included in the manuscript. – We have added that the planned study period is May 2018 - May 2020 in the methods section (p7)
- Due to the *nadA* gene variability, information about donor strain regarding to the *nadA* gene allele should be include. – The NadA gene sequence is derived from the Neisseria meningitidis MC58 strain which expresses allele 1. We have added this.
- *N. lactamica* or *N. meningitidis* detected on challenge volunteers throat swab or nasal wash taken at screening or at the pre-challenge visit is considered an exclusion criterion in the manuscript while in the study documents approved by DEFRA the exclusion criterion considered is pre-existing carriage of *Neisseria spp* (assessed 7 days prior to inoculation). – This is true but we argue that there is no difference in practical terms and the protocol described here is the one that was approved by the National Research Ethics Service.
- Why exclusion criteria 2, 5, 6 and 7 (Supplementary table 1) are only for challenge volunteers? The presence of these criteria in contact volunteers could affect their capability to acquisition of carriage

and therefore transmission results obtained from the study. – The reviewer is correct but DEFRA wanted us to collect information about transmission to `naïve` contacts in a real world setting so we decided that these exclusion criteria were sufficient to achieve that.

There are several references in the text not included in the “References” section (Harrison 2009, Evans 2011 . . .) – We have corrected this (p4 and p10)

VERSION 2 – REVIEW

REVIEWER	Caroline Vipond National institute for biological Standards and Control
REVIEW RETURNED	28-Jan-2019

GENERAL COMMENTS	An interesting study I look forward to reading the results of the trial, in particular the mucosal responses.
---

REVIEWER	Raquel Abad Instituto de Salud Carlos III. Spain
REVIEW RETURNED	11-Feb-2019

GENERAL COMMENTS	I appreciate that the suggestions and questions referred have been well addressed by the authors in the revised version.
--